# Perinatal Outcomes of Chronic Abruption Oligohydramnios Sequence: A Multicenter Retrospective Observational Study

**DOI:** 10.3390/jcm14155523

**Published:** 2025-08-05

**Authors:** Yoshifumi Kasuga, Yuka Fukuma, Kaoru Kajikawa, Keisuke Akita, Junko Tamai, Yuya Tanaka, Toshimitsu Otani, Marie Fukutake, Satoru Ikenoue, Mamoru Tanaka

**Affiliations:** Department of Obstetrics and Gynecology, Keio University School of Medicine, 5 Shinanomachi, Shinjuku-ku 160-8582, Tokyo, Japan

**Keywords:** chorioamnionitis, chronic abruption oligohydramnios sequence, miscarriage, preterm delivery

## Abstract

**Objective**: This study aimed to describe the perinatal and neonatal outcomes of chronic abruption oligohydramnios sequence in the Kanto region of Japan. **Methods**: This survey was conducted at 123 perinatal centers affiliated to this area. Data on the experience of managing chronic abruption oligohydramnios sequence between 1 January 2017, and 31 December 2022, were collected and analyzed. **Results**: Among the 82 cases of chronic abruption oligohydramnios sequence that were included in this study, there were seven miscarriages, five artificial abortions, and 70 deliveries beyond 22 gestational weeks (singleton: 68; twin: 2). In 82 patients, vaginal bleeding was the initial symptom of chronic abruption oligohydramnios sequence (88%). The mean gestational duration at the initial symptom onset was 17.3 ± 5.0 weeks. Of the 68 singleton pregnancies delivered after 22 gestational weeks, the mean gestational duration at delivery was 25.2 ± 2.8 weeks. In patients with chronic abruption oligohydramnios sequence, the mean white blood cell count at diagnosis and mean of the maximum white blood cell count during pregnancy were 11,589 ± 2885 and 15,357 ± 4745/μL, respectively; and the mean C-reactive protein at diagnosis and mean of the maximum C-reactive protein during pregnancy were 1.0 ± 1.2 and 2.0 ± 2.1 mg/L, respectively. Chorioamnionitis was identified in 43 patients (63%). All neonates were admitted to the neonatal intensive care unit. Of the 68 singleton neonates, 5 died immediately after birth. **Conclusions**: Chronic abruption oligohydramnios sequence is a rare perinatal complication that is possibly associated with infections, such as chorioamnionitis, and linked to adverse perinatal and neonatal outcomes.

## 1. Introduction

Worldwide, 13 million babies are born preterm (gestational age at birth <37 weeks) annually [1]. This number is approximately one in ten babies, and as preterm infants require a variety of medical interventions in the short and long term, reducing preterm births is a pressing issue worldwide. In addition, the long-term health of preterm infants may increase the risk of various diseases such as metabolic syndrome, obesity, delayed mental development, and psychiatric disorders [2,3,4,5,6,7]. This phenomenon is known as the Developmental Origins of Health and Disease (DOHaD). In addition, preventing preterm birth may have both short-term and long-term positive effects on health economics. However, preterm births have diverse causes. For example, maternal factors include nonpregnant thinness (body mass index [BMI] < 18.5 kg/m^2^), poor gestational weight gain, short stature, smoking, dental disease, and cervical incompetence, whereas perinatal complications include gestational hypertension, placenta previa, vas previa, and chorioamnionitis [8]. Fetal factors include fetal growth restriction and multiple pregnancies. Furthermore, preterm birth can be classified into three groups as follows: spontaneous with cervical dilation and uterine contraction, birth preceded by preterm rupture of membranes, and iatrogenic for any maternal or fetal problem [8]. In England, the rate of spontaneous preterm birth was 30–78% across the causes of preterm birth [9]. No established method for preventing preterm birth exists, partly owing to the variety of causes. However, as cases of idiopathic preterm birth continue to occur, ascertaining their etiology is important to prevent preterm birth. Jardine et al. strongly recommended that the etiology of preterm birth be evaluated in each patient and the cause of preterm birth be researched more strictly [10].

In this study, we focus on chronic abruption oligohydramnios sequence (CAOS) as a cause of preterm birth. First reported by Elliott et al. as a perinatal complication of abruptio placentae [11], CAOS is characterized by vaginal bleeding without placenta previa or other identifiable sources of bleeding, where the amniotic fluid volume is initially normal but later progresses to oligohydramnios (amniotic fluid index ≤ 5) without rupture of membranes [11]. CAOS is associated with a high risk for extreme preterm delivery [11,12,13], confers high fetal and neonatal mortality, and is the perinatal complication with the poorest prognosis [12,13,14]. However, only single-institution reports with small sample sizes (<20 cases) have been published to date, and the perinatal and neonatal outcomes of CAOS remain relatively unknown. Furthermore, since Elliott et al. reported CAOS, no further research about CAOS has been published in countries other than Japan. The preterm birth rate in Japan is lower than that in other developed countries [15]. Although the reason for the lower rate of preterm birth in Japan remains unknown, the frequent antenatal checkups that are conducted might contribute to reducing preterm birth. Clinicians potentially diagnose CAOS in Japan because of frequent antenatal checkups. Conversely, in other countries, miscarriages and premature births are expected to occur before CAOS is diagnosed. Therefore, having large-sample-size data on CAOS from Japan would be significant.

Against this background, in this multi-institutional study conducted in the Kanto region of Japan, we aimed to investigate the perinatal and neonatal outcomes of CAOS.

## 2. Materials and Methods

This study was a retrospective cohort study. This survey was performed at perinatal centers affiliated with the Kanto Society of Obstetrics and Gynecology (Appendix A). This region includes 123 perinatal centers, across 10 prefectures (i.e., Ibaraki, Tochigi, Gunma, Saitama, Chiba, Tokyo, Kanagawa, Yamanashi, Nagano, and Shizuoka), with a total population of 50 million; in 2022, 302,595 babies were born. This survey was performed between 26 June 2023, and 31 October 2023. We sent multiple reminders to potential participants to enhance the response rate. Between 1 January 2017, and 31 December 2022, we collected detailed information on patients with CAOS from every facility to elicit the experience of managing CAOS. Clinicians retrospectively collected maternal characteristics, blood test results, CAOS symptoms, and perinatal and neonatal outcomes from medical records. The diagnostic criteria for CAOS, as defined by Elliott et al., are as follows: (1) vaginal bleeding instead of an abnormal placenta position; (2) initially normal amniotic fluid volume; and (3) oligohydramnios without rupture [11]. Patients without oligohydramnios were excluded from the study, as were three deliveries beyond 22 weeks of gestation with unclear prognosis due to transfer to other facilities. The patients received perinatal management in each institution according to the Japan Society of Obstetrics and Gynecology and Japan Association of Obstetricians and Gynecologists guidelines [16]. However, as CAOS management was not included in these guidelines, each clinician provided care on a case-by-case basis. Clinical chorioamnionitis (CAM) was diagnosed according to the criteria specified in the guidelines issued by the Japan Society of Obstetrics and Gynecology and Japan Association of Obstetricians and Gynecologists [16,17]. Furthermore, according to Blanc’s classification [18], in all participating institutions, a pathological diagnosis of CAM was based on the presence of acute inflammatory lesions in the decidua, chorion, or amnion. Diffuse chorionic hemosiderosis (DCH) is diagnosed by diffuse hemosiderin-laden macrophages at the amnion of the free membrane and chorionic plate [19]. Given the varying perspectives on CAOS management, patients received perinatal care according to the institutional protocol. Fetal growth restriction (FGR) was defined as an estimated fetal body weight less than 1.5 standard deviations (SD), which corresponds to less than the 7th percentile [16]. Continuous values are presented as the mean ± SD, whereas categorical values are presented as the frequency (proportion). To compare the miscarriage group and the group with delivery after 22 weeks of gestation, we used Student’s *t*-test and the chi-square test. Statistical analysis was conducted using JMP Pro 17 (SAS Institute, Cary, NC, USA), with statistical significance set at *p* < 0.05.

This study was conducted with the approval of the Keio University School of Medicine (approval number: 20221197; approval date: 28 March 2023). The requirement of informed consent was waived owing to the retrospective study design.

## 3. Results

Responses were received from 70 (57%) facilities, of which 26 had implemented CAOS management. Among the 82 CAOS cases included in this study, there were seven miscarriages, five artificial abortions, and 70 deliveries after 22 weeks of gestation. Among the seven miscarriages and five artificial abortions (four primiparas), the mean gestational age at miscarriage and artificial abortion was 20.0 ± 1.3 and 20.4 ± 0.9 weeks, respectively. Of the 70 deliveries after 22 weeks, 2 were twin pregnancies, delivered by cesarean section (CS) at 25 and 28 weeks.

The characteristics of the 82 participants are listed in Table 1. The mean maternal age was 33.2 ± 5.8 years; 49% of participants were nulliparous. Of the participants, 76% conceived naturally, 9% via intrauterine insemination, 15% via in vitro fertilization and embryo transfer (IVF-ET), and 1% via oocyte donation. The most common CAOS initial symptom was vaginal bleeding (88%). Other detected initial symptoms included subchorionic hematoma (SCH), abdominal pain, FGR, and oligohydramnios. The mean gestational age at initial symptom onset was 17.3 ± 5.0 weeks. Table 2 compares maternal characteristics between the miscarriage group and the group with delivery after 22 gestational weeks. There were no significant between-group differences in maternal age and incidence of nulliparity. The IVF-ET incidence in the miscarriage group was significantly higher than that in the delivery group (57% vs. 9%, *p* = 0.004). In all miscarriages, the initial symptom was vaginal bleeding. The gestational age at initial symptom onset in the miscarriage group was significantly shorter than that in the group with delivery after 22 gestational weeks (13.3 ± 3.5 vs. 18.0 ± 4.9, *p* = 0.01).

Table 3 presents perinatal and placental findings for singleton pregnancies in women who delivered after 22 weeks. The mean duration from initial symptom to delivery was 7.1 ± 5.5 weeks. Many participants delivered within 10 weeks of initial symptoms (Figure 1). Of the 68 participants, 15 had intrapartum fever. The mean white blood cell (WBC) count at CAOS diagnosis was 11,589 ± 2885/μL, and the mean maximum WBC count during pregnancy was 15,357 ± 4745/μL. The mean C-reactive protein (CRP) at CAOS diagnosis was 1.0 ± 1.2 mg/L, with a mean maximum CRP of 2.0 ± 2.1 mg/L. The mean gestational age at delivery was 25.2 ± 2.8 weeks; 2 participants delivered at 37 weeks, whereas 47 delivered at or before 25 weeks. The gestational weeks at delivery (24.9 ± 2.5 vs. 26.7 ± 3.9, *p* = 0.07) and the incidence of delivery at or before 25 weeks (71% vs. 60%, *p* = 0.49) did not differ significantly between the groups with and without vaginal bleeding as the initial symptom. No delivery before 24 weeks occurred in women without vaginal bleeding as an initial symptom. The modes of delivery were CS and vaginal delivery in 48 and 20 cases, respectively (vaginal deliveries included four intrauterine fetal deaths). Among the 48 CS cases, the reasons for CS included non-reassuring fetal status (n = 22), acute placental abruption (n = 6), CAM (n = 6), and inability to control labor onset (n = 24). The median Apgar score at 1 min was 3 points (range: 0–8), whereas the Apgar score at 5 min was 6 points (0–10). The mean placental weight was 293 ± 111 g. Among the 68 cases, 65 were evaluated for placental pathophysiology; of these, we identified CAM and DCH in 43 (63%) and 22 (32%) cases, respectively.

Table 4 presents neonatal complications (duplicate data). All neonates were admitted to the neonatal intensive care unit. Of the 68 neonates, 5 died immediately after birth. Among them, 2 had sepsis, 34 had chronic lung disease (CLD), 5 had persistent pulmonary hypertension of the newborn, 13 had cerebral hemorrhage (including intraventricular hemorrhage [IVH]), and 2 had gastrointestinal perforation.

## 4. Discussion

The results showed that CAOS was a rare perinatal complication with poor prognosis. In this study that was conducted to elucidate the clinical characteristics of CAOS, we showed that early-gestational initial symptoms and IVF-ET are potential risk factors for miscarriage in patients with CAOS. The gestational duration at delivery was extremely short (25.2 ± 2.8 weeks). Particularly, patients with vaginal bleeding as the initial symptom tended to deliver earlier than those without vaginal bleeding (24.9 ± 2.5 vs. 26.7 ± 3.9 weeks, *p* = 0.07). Furthermore, high levels of maternal serum markers of infection were noted, and CAM was frequently detected. Therefore, if a patient develops CAOS, a facility that can provide advanced medical care should manage the patient.

Antenatal checkups in Japan are performed at medical institutions at least once a month (twice a month after the 24th gestational week and once a week after the 36th week) (published by the Japanese Ministry of Health, Labor, and Welfare: https://www.mhlw.go.jp/bunya/kodomo/boshi-hoken10/dl/02.pdf, accessed on 1 February 2025). Therefore, Japanese mothers receive antenatal checkups at least 15 times by the time of their deliveries at term. However, the number of antenatal checkups is fewer than 10 in many developed countries. Furthermore, access to hospitals is easy in Japan because all people can use the medical insurance system, and many hospitals and clinics exist. Especially in the first trimester, where initial symptoms of CAOS are detected, whether pregnant women can visit hospitals easily is very important for CAOS diagnosis. Owing to these conditions, most reports about CAOS have been published from Japan [20].

In the present study, pregnant women with initial symptoms during early gestation and those who conceived via IVF-ET were at a higher risk of miscarriage in CAOS. The initial symptoms of CAOS included not only vaginal bleeding but also SCH, abdominal pain, FGR, and oligohydramnios. Although the gestational duration at the time of occurrence of the initial vaginal bleeding has been previously evaluated [11,12,14], our investigation considered initial symptoms in addition to vaginal bleeding. A previous report identified a duration of vaginal bleeding >11 weeks as an important risk factor for CAOS in patients with preterm delivery [12]. In the present study, we did not collect data on the duration of vaginal bleeding because it rarely occurs continuously and confers difficulty in accurately measuring its duration. However, all miscarriages involved vaginal bleeding as an initial symptom. Furthermore, of the deliveries that occurred after 22 gestational weeks, the gestational duration at delivery tended to be shorter in those with, than in those without, vaginal bleeding as the initial symptom. This suggested that vaginal bleeding might be an important symptom in CAOS.

The mean gestational duration at delivery was 25.2 ± 2.8 weeks, which was the same as the duration mentioned in previous reports [11,12,13,14]. Therefore, CAOS is a potential risk factor for preterm delivery. Elliott et al. considered chronic abruptio placentae as the main cause of CAOS [11]. However, although magnetic resonance imaging (MRI) findings of patients with CAOS show evidence of intrauterine hematomas and hemorrhagic amniotic fluid, retroplacental hematoma was not detected pathologically and fetal membranes were extremely fragile and ragged. Therefore, Chigusa et al. considered that the biggest risk factor of CAOS might not be chronic abruption of the placenta but rather rupture of membranes [14]. However, in this study, we could not obtain data on MRI findings because it is uncommon to use MRI to evaluate CAOS in Japan. Moreover, in the present study, the incidence of pathological CAM was 63%, and 15 mothers had fever at delivery. Furthermore, high maternal WBC and CRP levels were detected prior to delivery in the participants, especially a maximum WBC >15,000/µL during pregnancy, which was observed in approximately half of the cohort (49%). Thus, an increased WBC count is an important clinical characteristic of CAM [17], and also suggests the possible association of infection with the development of CAOS. In previous reports, the incidence of pathological CAM was approximately 50% [12,14]. Acute placental abruption with CAM is associated with a higher risk of oligohydramnios, maternal postpartum fever, longer maternal hospitalization, and increased composite adverse neonatal morbidity than cases without CAM [21]. Therefore, infection may be a key factor in the pathogenesis of CAOS. However, the question remains as to which target bacteria should be selected to control CAM.

The most common microorganisms detected in clinical CAM at term without CAOS are the *Ureaplasma* species, and polymicrobial infections occurred in 70% of cases [22]. Moreover, the *Ureaplasma* species is associated with preterm deliveries [23,24]. In this study, we collected results from cultures of vaginal discharge samples obtained before delivery, revealing species such as *Gardnerella vaginalis*, *Escherichia coli*, and *Pseudomonas aeruginosa*. However, in Japan, it is uncommon to test for *Ureaplasma* species using vaginal discharge samples. Therefore, we could not investigate the association between CAOS and *Ureaplasma* species. In Japan, concurrent ampicillin and erythromycin administration is frequently used to control CAM. However, *Ureaplasma* species are often resistant to ampicillin and erythromycin [25,26,27]. Azithromycin and clarithromycin are useful for managing clinical CAM caused by the *Ureaplasma* species. Although we did not collect data on antibiotic use in the present study, azithromycin and clarithromycin use is unusual for CAM management in Japan. Further studies are required to determine whether infection with the *Ureaplasma* species is associated with CAOS, azithromycin, or clarithromycin.

DCH is frequently detected in patients with CAOS, and is associated with preterm birth, pulmonary hypertension of the newborn, and dry lung syndrome [13]. Chigusa et al. reported that DCH was detected in all cases with CAOS [14], and Kobayashi et al. reported that 53% of their CAOS patients had DCH [12]. Iizuka et al. reported that DCH was the main cause of oligohydramnios in three cases of CAOS [28]. The severity of DCH is related to amniotic epithelial necrosis, and the amniotic epithelium sustains DCH-associated oxidative stress. However, in this study, DCH was detected in 22 patients who delivered after 22 weeks of gestation (32%). As pathophysiological diagnosis was performed at each facility, uncertainty in diagnosing DCH might have reduced the incidence compared to other reports. However, we considered another reason, as suggested by Chigusa et al., for oligohydramnios, such as membrane rupture [14]. Therefore, CAM may be correlated with CAOS.

Neonates born to mothers with CAOS have a higher risk for CLD because they may aspirate bloody amniotic fluid and develop lung damage. Although frequent therapeutic amnioinfusions have not been successful in preventing preterm delivery, they could prevent lung injury [29]. Oligohydramnios is a risk factor for pulmonary hypoplasia. Lung damage from intrauterine causes might be the main cause of high neonatal mortality in CAOS. Cerebral hemorrhage, including IVH, was frequently detected in the present study. Although this may have been attributed to extreme prematurity at birth, CAM was associated with IVH in neonates born before 34 weeks of gestation [30].

This was a retrospective observational study, and the response rate was not very high; nonetheless, to our knowledge, this is the largest study to date to have investigated the perinatal and neonatal outcomes of CAOS. Previous reports included only single-institution data, and this is the first report to include data from multiple institutions. Furthermore, maternal serum markers of infection (WBC count and CRP) in blood samples have not been evaluated in previous studies. Therefore, we believe that this report is useful for providing crucial insights for clinicians undertaking the management of CAOS. However, owing to the retrospective cohort study design, we could not undertake a detailed evaluation of the mechanisms of CAOS, including MRI findings. We were unable to assess the other risk factors for preterm birth (e.g., smoking, body mass index, cervical insufficiency, etc.). Furthermore, as CAOS is a rare disease, no standard CAOS management protocol is currently available in Japan, and the medical care options available for patients with CAOS differ across regions and medical institutions. Therefore, further research, such as a prospective cohort study, is required to investigate the underlying mechanisms of CAOS. Based on our results, some patients underwent artificial abortions because of CAOS; however, we did not ascertain the detailed reasons for the selection of artificial abortion by the patients and their families. Nonetheless, as the institutions that participated in the present study were secondary- or tertiary-care facilities and CAOS might have developed in early gestation, more patients with CAOS might have selected artificial abortion for CAOS management at primary health centers.

## 5. Conclusions

CAOS is a rare complication that is linked to adverse perinatal and neonatal outcomes and may be associated with infections, such as CAM. This study comprises the largest regional dataset on CAOS in Japan, and highlighted the strong association of CAOS with preterm delivery, infection, and neonatal morbidity or mortality, all of which emphasize the need for early recognition and specialized care.

## Figures and Tables

**Figure 1 jcm-14-05523-f001:**
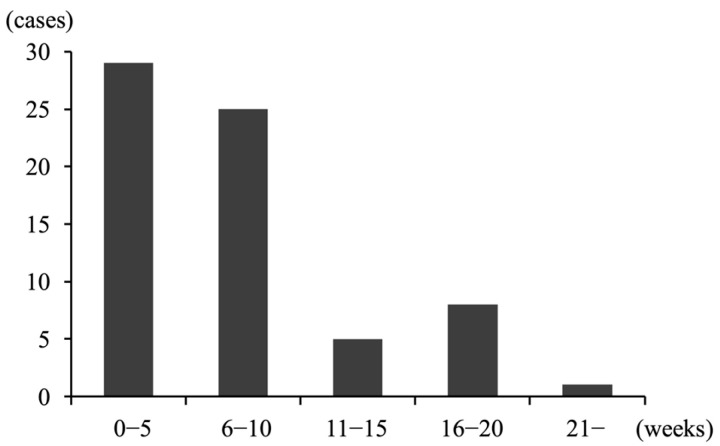
The number of cases stratified by the duration between initial symptom appearance and delivery in patients with chronic abruption oligohydramnios sequence who delivered after 22 weeks of gestation, of which many delivered within 10 weeks of initial symptom onset.

**Table 1 jcm-14-05523-t001:** Maternal characteristics of 82 cases with chronic abruption oligohydramnios sequence (CAOS).

Characteristics	N (%) or Mean (SD)
Maternal age (years)	33.2 ± 5.8
Nulliparous	40 (49%)
Mode of conception	
Natural intercourse	62 (76%)
Intrauterine insemination	7 (9%)
In vitro fertilization and embryo transfer	12 (15%)
Oocyte donation	1 (1%)
Initial symptoms and signs of CAOS	
Vaginal bleeding	72 (88%)
Subchorionic hematoma	3 (4%)
Abdominal pain	4 (5%)
Fetal growth restriction	3 (4%)
Oligohydramnios	6 (7%)
Gestational weeks at initial symptom (weeks)	17.3 ± 5.0
Outcome	
Miscarriage	7 (9%)
Artificial abortion	5 (6%)
Delivery after 22 gestational weeks	70 (85%)

Initial symptoms and signs total more than 100% because some patients had more than one finding.

**Table 2 jcm-14-05523-t002:** Comparison of characteristics between miscarriage group and group with delivery after 22 gestational weeks.

	Miscarriage (n = 7)	Delivery After 22 Gestational Weeks (n = 70)	*p*-Value
Maternal age	33.6 ± 5.2	33.0 ± 6.0	0.80
Nulliparity	4 (57%)	35 (50%)	1
In vitro fertilization and embryo transfer	4 (57%)	6 (9%)	0.004
Initial symptom and sign of CAOS			
Vaginal bleeding	7 (100%)	60 (86%)	
Subchorionic hematoma	0 (0%)	3 (5%)	
Abdominal pain	0 (0%)	2 (3%)	
Fetal growth restriction	0 (0%)	3 (5%)	
Oligohydramnios	0 (0%)	6 (10%)	
Gestational weeks at initial symptom	13.3 ± 3.5	18.0 ± 4.9	0.01

CAOS, chronic abruption oligohydramnios sequence. Initial symptoms and signs total more than 100% because some patients had more than one finding. Data are N (%) or mean ± SD.

**Table 3 jcm-14-05523-t003:** Perinatal and placental characteristics of singleton pregnancies in women who delivered after 22 gestational weeks.

Characteristics	Participants and Values
Duration between initial symptom and delivery (weeks)	7.1 ± 5.5
Maternal fever at delivery	15 (22%)
White blood cell count (µ/L)	
At diagnosis of CAOS	11,589 ± 2885
Maximum count during pregnancy	15,357 ± 4745
Participants with white blood cell count >15,000	33 (49%)
Maternal serum c-reactive protein (mg/L)	
At diagnosis of CAOS	1.0 ± 1.2
Maximum during pregnancy	2.0 ± 2.1
Gestational weeks at delivery	25.2 ± 2.8
Mode of delivery	
Vaginal delivery	20 (29%)
Cesarean section	48 (71%)
Apgar score at 1 min	3 (0–8)
Apgar score at 5 min	6 (0–10)
Placental weight (g)	293 ± 111
Placental pathophysiological findings	
Chorioamnionitis	43 (63%)
Diffuse chorionic hemosiderosis	22 (32%)
Unknown	3 (4%)

SD, standard deviation; CAOS, chronic abruption oligohydramnios sequence. Placental pathophysiological findings total more than 100% because some patients had more than one finding. Data are N (%), mean ± SD, or median (range).

**Table 4 jcm-14-05523-t004:** Neonatal complications.

Sepsis	2 (3%)
Respiratory distress syndrome	17 (27%)
Chronic lung disease	34 (50%)
Persistent pulmonary hypertension of the newborn	5 (7%)
Cerebral hemorrhage including intraventricular hemorrhage	13 (19%)
Patent ductus arteriosus	6 (9%)
Patent foramen ovale	1 (1%)
Gastrointestinal perforation	2 (3%)
Retinopathy of prematurity	18 (26%)

Neonatal complications total more than 100% because some patients had more than one finding.

## Data Availability

The data presented in this study are available upon request from the corresponding authors.

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
