# Peer review of "Perinatal Outcomes of Chronic Abruption Oligohydramnios Sequence: A Multicenter Retrospective Observational Study"

_jcm, 2025, doi:10.3390/jcm14155523_

Round 1
Reviewer 1 Report
Comments and Suggestions for Authors
A lot of references are missing in the introduction section. Lines 39-43 need references on DOHAD and risk for metabolic disorders, obesity, tumors etc. Lines 45-48 need references.
In lines 66-68, specify the sample sizes from previous publications (atleast a range). 'Small' sample sizes is vague.
This study itself has sample size of 82. Is it possible to include more data to increase sample size?
What is DCH?
Author Response
A lot of references are missing in the introduction section. Lines 39-43 need references on DOHAD and risk for metabolic disorders, obesity, tumors etc. Lines 45-48 need references.
Response: Thank you for your advice. We have added the following references to ensure that the factual statements are supported by relevant references.
- Song, P.; Hui, H.; Yang, M.; Lai, P.; Ye, Y.; Liu, Y.; Liu, X. Birth weight is associated with obesity and T2DM in adulthood among Chinese women. BMC Endocr. Disord. 2022, 22, 285, doi:10.1186/s12902-022-01194-1.
- Zanetti, D.; Tikkanen, E.; Gustafsson, S.; Priest, J.R.; Burgess, S.; Ingelsson, E. Birthweight, Type 2 Diabetes Mellitus, and Cardiovascular Disease: Addressing the Barker Hypothesis With Mendelian Randomization. Circ Genom Precis Med 2018, 11, e002054, doi:10.1161/CIRCGEN.117.002054.
- Schieve, L.A.; Tian, L.H.; Rankin, K.; Kogan, M.D.; Yeargin-Allsopp, M.; Visser, S.; Rosenberg, D. Population impact of preterm birth and low birth weight on developmental disabilities in US children. Ann. Epidemiol. 2016, 26, 267-274, doi:10.1016/j.annepidem.2016.02.012.
- Breslau, N.; Paneth, N.S.; Lucia, V.C. The lingering academic deficits of low birth weight children. Pediatrics 2004, 114, 1035-1040, doi:10.1542/peds.2004-0069.
- Barker, D.J.; Winter, P.D.; Osmond, C.; Margetts, B.; Simmonds, S.J. Weight in infancy and death from ischaemic heart disease. Lancet 1989, 2, 577-580.
- Boney, C.M.; Verma, A.; Tucker, R.; Vohr, B.R. Metabolic syndrome in childhood: association with birth weight, maternal obesity, and gestational diabetes mellitus. Pediatrics 2005, 115, e290-296, doi:10.1542/peds.2004-1808.
In lines 66-68, specify the sample sizes from previous publications (atleast a range). 'Small' sample sizes is vague.
Response: Thank you for pointing this out. Accordingly, we have revised the following sentence.
“However, only single-institution reports with small sample sizes (<20 cases) have been published to date, and the perinatal and neonatal outcomes of CAOS remain relatively unknown.” (p2, lines 65-66)
This study itself has sample size of 82. Is it possible to include more data to increase sample size?
Response: Thank you for your insightful question. However, we are unable to increase the sample size because this multicenter study was approved by the institutional ethics committee of our institution for performance during the specified study period, and 70 institutions participated in this retrospective study. Owing to these factors, additional data collection, verification, and analysis at this stage would prove challenging.
What is DCH?
Response: Thank you for pointing this out. We agree that the definition of the diagnostic criteria of DCH should have been described in the Method section. Therefore, we have added the following sentence and cited a relevant reference.
“Diffuse chorionic hemosiderosis (DCH) is diagnosed based on the presence of diffuse hemosiderin-laden macrophages at the amnion interface of the free membrane and the chorionic plate [19].” (p3, lines 103-105)
- Redline, R.W.; Wilson-Costello, D. Chronic peripheral separation of placenta. The significance of diffuse chorioamnionic hemosiderosis. Am. J. Clin. Pathol. 1999, 111, 804-810, doi:10.1093/ajcp/111.6.804.
Reviewer 2 Report
Comments and Suggestions for Authors
In line 90 and 249-250: In this manuscript the diagnostic criteria for CAOS 3. adopt oligohydramnios without rupture and ref 8. Chigusa et al: Chronic abruption-oligohydramnios sequence (CAOS) revisited: possible implication of premature rupture of membranes. However, In this manuscript the oligohydramnios may result in continuous minimal of amniofluid leakage and you do not mention the methods to prove it reality without abruption(Leakage).
Author Response
In line 90 and 249-250: In this manuscript the diagnostic criteria for CAOS 3. adopt oligohydramnios without rupture and ref 8. Chigusa et al: Chronic abruption-oligohydramnios sequence (CAOS) revisited: possible implication of premature rupture of membranes. However, In this manuscript the oligohydramnios may result in continuous minimal of amniofluid leakage and you do not mention the methods to prove it reality without abruption (Leakage).
Response: Thank you for your insightful question. We agree that this is a very important point regarding any discussion of CAOS. However, Elliot et al.’s definition of CAOS is currently accepted, and has not been changed yet. Therefore, CAOS continues to be diagnosed on the basis of the definition suggested by Elliot. However, the etiopathogenesis of COAS remains unclear. Chigusa et al. proposed that amniotic fluid leakage or rupture of membranes might be associated with CAOS. Based on our study data, we cannot discuss the detailed pathogenic mechanisms underlying CAOS, as this should be elucidated in further research. Accordingly, we have mentioned this limitation in the Discussion:
“However, owing to the retrospective cohort study design, we could not undertake a detailed evaluation of the mechanisms of CAOS, including MRI findings. Furthermore, as CAOS is a rare disease, no standard CAOS management protocol is currently available in Japan, and the medical care options available for patients with CAOS differ across regions and medical institutions. Therefore, further research, such as a prospective cohort study, is required to investigate the underlying mechanisms of CAOS.” (p 8, lines 280-285)
Reviewer 3 Report
Comments and Suggestions for Authors
Comments to the Authors:
This multicenter retrospective observational study investigates the perinatal and neonatal outcomes of chronic abruption oligohydramnios sequence (CAOS) in the Kanto region of Japan. The study included 82 CAOS cases from 123 perinatal centers, analyzing maternal characteristics, clinical symptoms, infection markers, and neonatal complications. The conclusion states that hronic abruption oligohydramnios sequence is a rare perinatal complication linked to adverse perinatal and neonatal outcomes and infections like 30 chorioamnionitis. This study demonstrates some clinical significance, but the lack of scientific robustness makes it unsuitable for publication in the present version.
Major points:
- The retrospective design of the study protocol carries risks of selection bias, exemplified by a facility response rate of only 57%, coupled with the absence of standardized CAOS management protocols across participating centers;
- The presented research data exhibits incompleteness, particularly the lack of MRI examinations or microbiological analyses to investigate the etiology of CAM. We recommend supplementing these critical components;
- Regarding the conclusions, the assertions about causal relationships with CAM lack supporting microbiological evidence. We suggest incorporating additional literature references to substantiate these claims.

Comments on the Quality of English Language
The manuscript contains instances of incorrect tense and voice usage. For example, in the Introduction section (Page 2), "research...have been published" should be revised to "research...has been published" . It is recommended to thoroughly review the entire manuscript and implement consistent corrections.
Author Response
Comments to the Authors:
This multicenter retrospective observational study investigates the perinatal and neonatal outcomes of chronic abruption oligohydramnios sequence (CAOS) in the Kanto region of Japan. The study included 82 CAOS cases from 123 perinatal centers, analyzing maternal characteristics, clinical symptoms, infection markers, and neonatal complications. The conclusion states that chronic abruption oligohydramnios sequence is a rare perinatal complication linked to adverse perinatal and neonatal outcomes and infections like 30 chorioamnionitis. This study demonstrates some clinical significance, but the lack of scientific robustness makes it unsuitable for publication in the present version.
Major points:
- The retrospective design of the study protocol carries risks of selection bias, exemplified by a facility response rate of only 57%, coupled with the absence of standardized CAOS management protocols across participating centers;
Response: Thank you for the valuable feedback. We completely agree that the response rate was not as high as it should have been, and we have described this limitation in the Discussion. Although we repeatedly sent reminders to every institution for participation in this study, some clinicians did not respond. Moreover, this was a retrospective cohort study. Furthermore, as CAOS is a rare disease, no standard management protocol is currently available in Japan, and the medical care options available for patients with CAOS differ across regions and medical institutions. We have clarified these details in the revised manuscript as follows:
[Method]
“This survey was performed between June 26, 2023, and October 31, 2023. We sent repeated reminders to the potential participants to increase the response rate of this study.” (p 2, line 84-86)
[Discussion]
“However, owing to the retrospective cohort study design, we could not undertake a detailed evaluation of the mechanisms of CAOS, including MRI findings. Furthermore, as CAOS is a rare disease, no standard CAOS management protocol is currently available in Japan, and the medical care options available for patients with CAOS differ across regions and medical institutions. Therefore, further research, such as a prospective cohort study, is required to investigate the underlying mechanisms of CAOS.” (p 8, lines 280-285)
- The presented research data exhibits incompleteness, particularly the lack of MRI examinations or microbiological analyses to investigate the etiology of CAM. We recommend supplementing these critical components;
Response: Thank you for this comment. As described in the Discussion section, in Japan, MRIs are not routinely used to evaluate CAOS. We have clarified this aspect in the revised manuscript. Furthermore, we agree with your observation regarding the inadequate description of the diagnosis of CAM and have added the following sentences in the main text as well as included additional references.
[Methods]
“Clinical chorioamnionitis (CAM) was diagnosed according to the criteria specified in the guideline issued by the Japan Society of Obstetrics and Gynecology and Japan Association of Obstetricians and Gynecologists [10,11]. Furthermore, according to Blanc’s classification [12], in all participating institutions, a pathological diagnosis of CAM was based on the presence of acute inflammatory lesions in the decidua, chorion, or amnion.” (p 3, lines 98-103)
[Discussion]
“However, owing to the retrospective cohort study design, we could not undertake a detailed evaluation of the mechanisms of CAOS, including MRI findings. Furthermore, as CAOS is a rare disease, no standard CAOS management protocol is currently available in Japan, and the medical care options available for patients with CAOS differ across regions and medical institutions. Therefore, further research, such as a prospective cohort study, is required to investigate the underlying mechanisms of CAOS.” (p 8, lines 280-285)
[References]
- Tita, A.T.; Andrews, W.W. Diagnosis and management of clinical chorioamnionitis.Clin Perinatol 2010, 37, 339-354, https://doi.org/1016/j.clp.2010.02.003.
- Blanc, W.A. Pathology of the placenta, membranes, and umbilical cord in bacterial, fungal, and viral infections in man.Monogr Pathol 1981, 67-132.
Regarding the conclusions, the assertions about causal relationships with CAM lack supporting microbiological evidence. We suggest incorporating additional literature references to substantiate these claims.
Response: Thank you for your valuable advice. We agree that a detailed explanation of this point is required, and have accordingly added the following sentences and reference.
[Discussion]
“Furthermore, high maternal WBC and CRP levels were detected prior to delivery in the participants, especially the maximum WBC >15,000/µL during pregnancy that was observed in approximately half of the cohort (49%). Thus, an increased WBC count is an important clinical characteristic of CAM [11], and also suggests the possible association of infection with the development of CAOS.” (p 7, lines 228-232)
[References]
- Tita, A.T.; Andrews, W.W. Diagnosis and management of clinical chorioamnionitis.Clin Perinatol2010, 37, 339-354, https://doi.org/10.1016/j.clp.2010.02.003.
Round 2
Reviewer 1 Report
Comments and Suggestions for Authors
Comments addressed.
Author Response
Comments addressed.
Thank you for your reviewing our manuscript.
Reviewer 3 Report
Comments and Suggestions for Authors
Comments to the Authors:
This multicenter retrospective observational study investigates the perinatal and neonatal outcomes of chronic abruption oligohydramnios sequence (CAOS) in the Kanto region of Japan. The study included 82 CAOS cases from 123 perinatal centers, analyzing maternal characteristics, clinical symptoms, infection markers, and neonatal complications. The conclusion states that hronic abruption oligohydramnios sequence is a rare perinatal complication linked to adverse perinatal and neonatal outcomes and infections like 30 chorioamnionitis. This study demonstrates potential clinical relevance; however, further methodological refinement and more in-depth data analysis would strengthen its validity and impact.
Major points:
- Only 57% of institutions (70/123) responded to the survey, and the differences between responding and non-responding institutions were not analyzed. We recommend adding sensitivity analyses to assess potential bias effects on the results.
- Common preterm birth risk factors (e.g., smoking, BMI, cervical insufficiency) were not adjusted for, potentially overestimating CAOS-specific effects. We suggest including these known confounding factors in multivariate regression analyses.
- Tables 1 and 2 show duplicate counting of 'initial symptoms' (e.g., vaginal bleeding being counted both as a primary symptom and under other symptoms). Clarification is needed regarding whether multiple selections were permitted.
- Although a high incidence of CAM (63%) was observed, specific pathogens (e.g., Ureaplasma species) were not investigated.
Author Response
This multicenter retrospective observational study investigates the perinatal and neonatal outcomes of chronic abruption oligohydramnios sequence (CAOS) in the Kanto region of Japan. The study included 82 CAOS cases from 123 perinatal centers, analyzing maternal characteristics, clinical symptoms, infection markers, and neonatal complications. The conclusion states that hronic abruption oligohydramnios sequence is a rare perinatal complication linked to adverse perinatal and neonatal outcomes and infections like 30 chorioamnionitis. This study demonstrates potential clinical relevance; however, further methodological refinement and more in-depth data analysis would strengthen its validity and impact.
Major points:
- Only 57% of institutions (70/123) responded to the survey, and the differences between responding and non-responding institutions were not analyzed. We recommend adding sensitivity analyses to assess potential bias effects on the results.
Response: Thank you for your insightful question. We share your perspective. However, as you know, non-responding institutions did not provide any feedback. Therefore, due to the missing data, we were unable to assess potential bias.
- Common preterm birth risk factors (e.g., smoking, BMI, cervical insufficiency) were not adjusted for, potentially overestimating CAOS-specific effects. We suggest including these known confounding factors in multivariate regression analyses.
Response: Thank you for your editorial comment. As you pointed out, it is important to evaluate the causes of preterm birth and whether the patients had these factors. However, we cannot evaluate the risk factors for preterm birth in CAOS because we did not receive the following data (e.g., smoking, BMI, cervical insufficiency, etc.) in this study. To express this, we added the following sentence to the Discussion section.
“We were unable to assess the other risk factors for preterm birth (e.g., smoking, body mass index, cervical insufficiency, etc.).” (P 8, lines 279–281)
- Tables 1 and 2 show duplicate counting of 'initial symptoms' (e.g., vaginal bleeding being counted both as a primary symptom and under other symptoms). Clarification is needed regarding whether multiple selections were permitted.
Response: Thank you for your valuable question. As shown in Tables 1 and 2, the data allowed for duplication and multiple selections.
- Although a high incidence of CAM (63%) was observed, specific pathogens (e.g., Ureaplasma species) were not investigated.
Response: Thank you for your insightful question. This is a valuable point, and we wanted to understand the association between CAOS and CAM in relation to Ureaplasma species. However, in Japan, testing for Ureaplasma species is uncommon in cases of bacterial vaginosis. Therefore, we could not correct the data on Ureaplasma species. As described in the Discussion section (P 8, lines 249–250), further studies are required to determine whether infection with Ureaplasma species is associated with CAOS, azithromycin, or clarithromycin.